# The MEK-ERK-MST1 Axis Potentiates the Activation of the Extrinsic Apoptotic Pathway during GDC-0941 Treatment in Jurkat T Cells

**DOI:** 10.3390/cells8020191

**Published:** 2019-02-21

**Authors:** Jana Nováková, Pavel Talacko, Petr Novák, Karel Vališ

**Affiliations:** 1BIOCEV, Institute of Microbiology, v.v.i., The Czech Academy of Sciences, Průmyslová 595, 252 50 Vestec, Czech Republic; jana.novakova@biomed.cas.cz; 2Department of Biochemistry, Faculty of Science, Charles University, Hlavova 8, 128 43 Prague, Czech Republic; pavel.talacko@natur.cuni.cz

**Keywords:** Hippo/MST1, caspase, apoptosis, PI3K, AKT, MEK, ERK

## Abstract

The discrete activation of individual caspases is essential during T-cell development, activation, and apoptosis. Humans carrying nonfunctional caspase-8 and caspase-8 conditional knockout mice exhibit several defects in the progression of naive CD4^+^ T cells to the effector stage. MST1, a key kinase of the Hippo signaling pathway, is often presented as a substrate of caspases, and its cleavage by caspases potentiates its activity. Several studies have focused on the involvement of MST1 in caspase activation and also reported several defects in the immune system function caused by MST1 deficiency. Here, we show the rapid activation of the MEK-ERK-MST1 axis together with the cleavage and activation of caspase-3, -6, -7, -8, and -9 after PI3K signaling blockade by the selective inhibitor GDC-0941 in Jurkat T cells. We determined the phosphorylation pattern of MST1 using a phosphoproteomic approach and identified two amino acid residues phosphorylated in an ERK-dependent manner after GDC-0941 treatment together with a novel phosphorylation site at S21 residue, which was extensively phosphorylated in an ERK-independent manner during PI3K signaling blockade. Using caspase inhibitors and the inhibition of MST1 expression using siRNA, we identified an exclusive role of the MEK-ERK-MST1 axis in the activation of initiator caspase-8, which in turn activates executive caspase-3/-7 that finally potentiate MST1 proteolytic cleavage. This mechanism forms a positive feed-back loop that amplifies the activation of MST1 together with apoptotic response in Jurkat T cells during PI3K inhibition. Altogether, we propose a novel MEK-ERK-MST1-CASP8-CASP3/7 apoptotic pathway in Jurkat T cells and believe that the regulation of this pathway can open novel possibilities in systemic and cancer therapies.

## 1. Introduction

Programmed cell death (apoptosis) is a crucial process that is essential for the survival of nearly all eukaryotic organisms. Apoptosis drives the development of multicellular organisms, regulates immune system functions, and controls the disposal of damaged cells in tissues [1]. It also occurs in unicellular organisms exposed to a wide range of stresses [2]. Caspases form an important group of cysteine proteases that play important roles during apoptosis. These enzymes share conserved cysteine–histidine dyads in their structures, which catalyze the nucleophilic attack and cleavage of the peptide bond after recognizing aspartic acid residues in the target proteins [3]. The activation of cellular caspases is under strict control, which is realized by several signaling pathways together with activatory and inhibitory proteins [4]. Caspases can be activated by several mechanisms exemplified by recruitment-activation (the activation of caspase-8 by CD95 receptor ligation), trans-activation (caspase-3 or -7 activation by caspase-8 or -9), and auto-activation (caspase-3 activation by RGD peptides) [5,6]. Caspase activation usually results in the proteolytic degradation of crucial cellular structures followed by apoptotic cell death, but several non-apoptotic functions of caspases have also been documented.

MST1 kinase forms a part of the central kinase cassette of the Hippo signaling pathway, which acts as a strong tumor-suppressor due to its ability to induce apoptosis in eukaryotic cells [7]. MST1 undergoes proteolytic activation and negatively regulates the transcription of several anti-apoptotic genes (*BCL2*, *cIAP*s, or *BCL-xL*) mainly through YAP/TAZ co-activator inhibition and stimulates the transcription of several pro-apoptotic genes (*NOXA*, *PUMA*, or *FASL*) through the activation of transcription factors such as FOXOs [8,9,10]. Previous studies have demonstrated MST1 cleavage by caspases as a core mechanism of MST1 kinase activation under specific types of cellular stresses, followed by caspase activity amplification through MST1 kinase action [11,12,13]. On the other hand, several studies have focused on caspase activation by MST1 through the regulation of BIM protein activity or the phosphorylation and inhibition of BCL-xL protein [14,15,16]. These results have provided evidence for the caspase-independent activation of MST1, although the mechanism involved remains obscure [17,18]. Moreover, several signaling pathways can regulate MST1 activity. The AKT-mediated phosphorylation of MST1 at T120 and T387 residues results in MST1 inhibition as well as MST1 phosphorylation by mTORC2 at S438 residue [12,19]. Unfortunately, kinases responsible for the activatory phosphorylation of MST1 remain folded in mist, but the regulation of the Hippo pathway activity by the metabolic status and the regulation of cellular metabolism by MST1 have been recently demonstrated [20,21].

We confirmed the abrogation of AKT kinase phosphorylation at S473 followed by the fast activation of MST1 and caspase-3, -6, -7, -8, and -9 in a model of Jurkat T cells after treatment with the selective PI3K inhibitor GDC-0941. The inhibition of AKT kinase activity by GDC-0941 was confirmed as a decrease in the phosphorylation of downstream P70S6K kinase at S424 residue and increase in caspase-9 activity. We also observed an increase in the phosphorylation of MEK1/2 at S217/S221 residues and ERK1/2 kinase phosphorylation at T202/Y204 residues, although the phosphorylation of c-RAF at S338 residue was decreased. ERK1/2 kinase activation was confirmed by an increase in the phosphorylation of P90RSK at S380 residue which represents the established target of EKR1/2 signaling. Then, we determined the phosphorylation status of MST1 in control cells, GDC-0941-treated cells and cells treated with a combination of GDC-0941 and the ERK1/2 inhibitor, SCH-772984, with the aim to test the potential impact of MEK-ERK signaling on MST1 phosphorylation. We characterized several phosphorylated amino acid residues of MST1 and identified the ERK-dependent phosphorylation sites of MST1 at T177 and T387 residues together with a novel ERK-independent phosphorylation site at S21 residue. Using a bioinformatic approach, we predicted PLK1, NEK2 and GSK3B signaling to have a potential impact on MST1 phosphorylation. To clarify the role of individual caspases during the proteolytic activation of MST1, we investigated the effects of the selected caspase activity inhibitors on MST1 cleavage. Pre-treatment of cells with the Z-DEVD-FMK inhibitor completely blocked the proteolytic activation and activity of caspase-3, -6, -7, and -9 after GDC-0941 treatment. However, the Z-DEVD-FMK inhibitor was insufficient to block the proteolytic activation of caspase-8 and MST1 which were significantly blocked by the same concentration of the general Z-VAD-FMK inhibitor. Moreover, MST1 inhibition by siRNA significantly reduced the proteolytic activation of caspase-3, -7, and -8 together with proteolytic activity against the Ac-DEVD-pNA substrate during GDC-0941 treatment. These results demonstrate the MEK-ERK-MST1 axis as a novel regulator of the extrinsic apoptotic pathway in Jurkat T cells.

## 2. Results

### 2.1. Selective PI3K Inhibition Triggers the Proteolytic Activation of MST1

MST1 kinase phosphorylation by AKT kinase inhibits its activity; therefore, we tested the effect of the selective PI3K inhibitor GDC-0941 on the phosphorylation levels and activity of these two kinases. We observed a significant decrease in AKT phosphorylation at S473 residue together with the proteolytic activation of MST1 after 2 h of GDC-0941 treatment (Figure 1A). During the search for changes in the phosphorylation of other kinases associated with the PI3K-AKT pathway, we observed a decrease in the phosphorylation of P70S6K at S424 and c-RAF at S338 together with an increase in the phosphorylation of MEK1/2 kinase at S217/S221, ERK1/2 kinase at T202/Y204 and P90RSK kinase at S380. 

P70S6K represents kinase downstream of AKT and a decrease in its phosphorylation is a marker of the inhibition of AKT activity by GDC-0941. On the other hand, MEK1/2 represents the well-known target of c-RAF but its phosphorylation was increased, suggesting the non-canonical activation of MEK-ERK signaling during GDC-0941 treatment. The activation of MEK-ERK signaling during GDC-0941 treatment was confirmed by an increase in the phosphorylation of P90RSK, a well-established target of ERK1/2 (Figure 1B).

### 2.2. ERK1/2 Signaling Inhibition Decreases the Proteolytic Activation of MST1 during GDC-0941 Treatment

Since we observed an increase in ERK1/2 phosphorylation during GDC-0941 treatment in Jurkat T cells, we tested the potential effect of ERK1/2 on the proteolytic activation of MST1. Pre-treatment of cells with the selective inhibitor of ERK1/2 signaling SCH-772984 resulted in a significant decrease in the phosphorylation of c-RAF, ERK1/2 and P90RSK together with an increase in the phosphorylation of MEK1/2 after GDC-0941 treatment (Figure 1B). ERK1/2 signaling inhibition exerted an inhibitory effect on the proteolytic activation of MST1 during GDC-0941 treatment (Figure 1B). We did not observe any effect of the SCH-772984 treatment alone on AKT, P70S6K, MEK1/2, ERK1/2, P90RSK phosphorylation and MST1 cleavage (Figure 1B). These results suggest a mechanism involving P90RSK phosphorylation at S380 residue, which is alternative to ERK1/2 signaling in control Jurkat T cells.

### 2.3. GDC-0941 Treatment Stimulates the Phosphorylation of T177 and T387 Residues of MST1 in an ERK-Dependent Manner

To clarify the effect of PI3K, ERK1/2 signaling inhibition and their combination on MST1 phosphorylation, we performed MST1 pull-down in Jurkat T cells, followed by the identification of individual phosphorylated residues and quantification of the phosphorylation levels using mass spectrometry (MS). We were able to quantify the phosphorylation levels of eight individual AA residues of MST1 protein (S21, S40, T177, T340, T387, S410, S414, and S438). The phosphorylation of seven residues was previously identified and was described in the PhosphoSite database but phosphorylation at S21 was newly identified in our study (Figure 1C,D). The position of S21 is highly conserved in evolution, highlighting the potential function of S21 in the regulation of MST1 activity (Figure 2A). Moreover, we revealed that phosphorylation at T177 and T387 residues occurred in an ERK1/2-dependent manner and ERK1/2 was also shown to play a crucial role in the proteolytic activation of MST1.

### 2.4. PLK1, GSK3 and NEK2 Kinases Were Predicted as Regulators of MST1 Phosphorylation

We matched sequences around the phosphorylated sites of MST1 against the database of motifs recognized by individual kinases using the Eukaryotic Linear Motif (ELM) resource. This resource utilizes information on primary, secondary, and tertiary structures to determine the active regulatory motifs of individual proteins. We found that the sequence around S438 looks like a motif recognized by NEK2 kinase, the sequence around S21 looks like a motif recognized by PLK1 kinase, and that the sequence around S414 looks like a motif recognized by GSK3B kinase (Figure 2B–D). The direct interaction of MST1 with PLK1 and NEK2 was reported previously and the regulation of several Hippo pathway components by GSK3B was also described. Then, we searched for the potential links between kinases predicted by the ELM and MEK-ERK signaling pathway. We found that PLK1 and NEK2 were previously considered to be regulated by MEK1/2 and that GSK3B is known to be regulated by P90RSK. These results reveal MEK1/2-PLK1/NEK2-MST1 and P90RSK-GSK3B-MST1 as novel signaling circuits that can orchestrate MST1 protein activation in Jurkat T cells.

### 2.5. GDC-0941 Treatment Activates Caspases in Jurkat T cells

We determined the activity of individual caspases after 2 h of GDC-0941 treatment using the Ac-DEVD-pNA substrate which is specific to caspase-3/-7, the Ac-LEHD-pNA substrate which is specific to caspase-9, the Ac-VEID-pNA substrate which is specific to caspase-6, and the Ac-YVAD-pNA substrate which is specific to caspase-1. We found a six-fold increase in caspase-3/-7 activities toward the Ac-DEVD-pNA substrate, two-fold increase in caspase-9 activity toward the Ac-LEHD-pNA substrate, insignificant increase in caspase-6 activity toward the Ac-VEID-pNA substrate, and no effect on caspase-1 activity toward the Ac-YVAD-pNA substrate (Figure 3A). This observation suggests the potential trans-activation of executive caspase-3/-7 by initiator caspase-8/-9. To test the effect of caspase-3/-7 activity on the proteolytic activation of MST1, we blocked those using experimentally determined concentrations of the Z-DEVD-FMK inhibitor and the Z-VAD-FMK pan-caspase inhibitor. We observed a complete inhibition of proteolytic activity toward the Ac-DEVD-pNA, Ac-LEHD-pNA, and Ac-VEID-pNA substrates by 2 µmol/L Z-DEVD-FMK inhibitor and 1 µmol/L Z-VAD-FMK inhibitor (Figure 3B).

### 2.6. GDC-0941 Treatment Activates the Extrinsic Apoptotic Pathway

Using immunoblotting, we confirmed the generation of cleaved forms of caspase-3, -7, and, -8 and a decrease in the non-cleaved forms of caspase-3, -6, -7, and -8 after 2 h of GDC-0941 treatment. We could not detect the generation of the cleaved form of caspase-6 (Figure 4). Pre-treatment of cells with the caspase inhibitor Z-DEVD-FMK completely inhibited the generation of cleaved forms and decrease in the levels of the full-length forms of all caspases mentioned above except for caspase-8, the activation of which was significantly inhibited only by treatment with Z-VAD-FMK (Figure 4). These observations highlighted the activation of caspase-8 as a crucial mechanism responsible for apoptosis induction in Jurkat T cells during GDC-0941 treatment.

### 2.7. Activation of MST1 during GDC-0941 Treatment Is Less Sensitive to Caspase Inhibitors

We tested the effect of selected concentrations of individual caspase inhibitors on the proteolytic activation of MST1 by immunoblotting. We found that 2 µmol/L Z-DEVD-FMK, which effectively inhibited the generation of cleaved forms of caspase-3/-7 and decreased the levels of the full-length forms of caspase-3, -7, and -6, was insufficient to inhibit the generation of the cleaved form of MST1 and to decrease the levels of the full-length form of MST1. Significant inhibition of the proteolytic activation of MST1 during GDC-0941 treatment was achieved using a higher Z-DEVD-FMK concentration (5 µmol/L; Figure 5A). Notably, 2 µmol/L Z-VAD-FMK significantly inhibited levels of the cleaved MST1 but a decrease in the levels of full-length MST1 could be observed (Figure 5A). These results do not support the primary role of caspase-3/7 during the proteolytic activation of MST1 after GDC-0941 treatment.

### 2.8. Downregulation of MST1 Protein Levels Attenuates Activation of Caspases during GDC-0941 Treatment

To clarify the potential involvement of MST1 in caspase activation during GDC-0941 treatment, we inhibited MST1 levels in Jurkat T cells using transfection with MST1-specific and non-target (NT) control siRNA. We confirmed the inhibition of the proteolytic activation of caspase-3, -7, and -8 in cells transfected with MST1-specific siRNA compared with that in cells transfected with NT control siRNA after GDC-0941 treatment (Figure 5B). Moreover, we also observed significant reduction in proteolytic activity toward the Ac-DEVD-pNA substrate in cells transfected with MST1-specific siRNA compared with that in cells transfected with NT control siRNA (Figure 5C). These results suggest the activation of MST1 kinase prior to the proteolytic activation of caspase and highlight MST1 kinase as a regulator of caspase activity and apoptosis in Jurkat T cells.

## 3. Discussion

MST1 kinase was recently identified as a key regulator of apoptosis in immune cells [22,23,24]. In this study, we investigated the modes of the regulation of MST1 activity through PI3K signaling and focused on the potential interplay between MST1 and cellular caspases during apoptosis induction in Jurkat T cells. PI3K acts as a key regulator of the lipid signaling pathway, leading to PDK1 signaling activation, which in turn activates AKT kinase. AKT regulates several other signaling pathways that are essential for cellular proliferation and apoptosis inhibition. AKT can activate mTORC1 signaling, which acts as a positive regulator of P70S6K and protein synthesis [25]. It has also been described as an important regulator of RAF-MEK-ERK signaling and is thus considered as an important regulator of cellular metabolism and proliferation [26,27]. The AKT-mediated phosphorylation of the BAD protein triggers BAD sequestration by 14-3-3 and apoptosis inhibition [28]. AKT can also inhibit apoptosis through the regulation of several IAP proteins, leading to the inhibition of individual caspases [29]. In addition, AKT signaling inhibits the activation of pro-apoptotic kinase MST1, anti-proliferative kinase GSK3B, and initiator caspase-9 [30,31,32]. In this context, we analyzed the impact of the selective PI3K inhibitor GDC-0941 on the individual aspects of AKT signaling.

We confirmed the complete inhibition of AKT phosphorylation at S473 residue after 2 h of GDC-0941 treatment (Figure 1A). Consistent with the current model of AKT signaling, we observed a decrease in P70S6K phosphorylation at S424 residue, suggesting the inhibition of mTORC1 activity and the proteolytic activation of MST1 kinase (Figure 1A,B). We also confirmed an increase in caspase-9 activity using the Ac-LEHD-pNA substrate (Figure 3A). However, we observed a decrease in the phosphorylation of c-RAF at S338 residue. When we investigated the phosphorylation status of kinases downstream of c-RAF, we found an increase in the phosphorylation of MEK1/2 at S217/S221 residues together with an increase in ERK1/2 kinase phosphorylation at T202/Y204 residues during GDC-0941 treatment (Figure 1A,B). These results suggest the involvement of the non-canonical pathway responsible for MEK-ERK signaling activation after GDC-0941 treatment. Our phosphoproteomic study highlighted the potential activation of the mTORC2 complex during PI3K inhibition as MST1 phosphorylation at S438 residue was increased. Moreover, we could not detect any significant levels of the phosphorylated forms of ERK1/2 in control cells, which propose a novel circuit in the regulation of MEK-ERK signaling. To determine ERK1/2 activity in Jurkat T cells, we monitored the impact of GDC-0941, SCH-772984, and their combination on the phosphorylation status of P90RSK which is localized downstream of ERK1/2. Surprisingly, we observed significant levels of P90RSK phosphorylated at S380 residue in control cells and an increase in this phosphorylation after GDC-0941 treatment, reflecting an increase in ERK1/2 kinase activity (Figure 1B). Moreover, the ERK1/2 inhibitor SCH-772984 had no effect on P90RSK phosphorylation and a combination of GDC-0941 and SCH-772984 almost completely abrogated the P90RSK phosphorylation levels. These results highlight the alternative phosphorylation of P90RSK, which could be regulated by the PI3K pathway and the switch to phosphorylation through the MEK-ERK pathway after PI3K inhibition. The pathways responsible for the non-canonical activation of MEK-ERK signaling during GDC-0941 treatment remain unclear, although the RAF-independent activation of MEK-ERK signaling was reported previously [33].

Specific ERK1/2 signaling inhibition decreased the proteolytic activation of MST1 during GDC-0941 treatment in Jurkat T cells. To characterize the impact of PI3K and ERK signaling on MST1, we determined the phosphorylation levels of individual AA residues in control cells and cells treated with GDC-0941, SCH-772984, and their combination. Using MS analysis, we determined eight AA residues that change in the phosphorylation levels between the tested conditions (Figure 1C). Phosphorylation at T177 and T387 residues was significantly increased after GDC-0941 treatment and decreased after SCH-772984 treatment, whereas the effect of GDC-0941 was completely abolished by SCH-772984 pre-treatment, suggesting the ERK1/2-dependent phosphorylation of these residues. T177 and T387 residues are involved in the activation of MST1, but previous reports have suggested auto-phosphorylation (T177) and AKT-dependent phosphorylation (T387) as the mechanisms responsible [30,34]. However, our results demonstrated the dependency of these phosphorylation sites on ERK1/2 activity, especially after PI3K signaling inhibition. Moreover, the ELM resource predicted only one AKT-responsive phosphorylation site at the T120 residue of MST1 [35]. Phosphorylation at T387 residue was described to participate in MST1 binding to MOB1, which regulates MST1 activity toward specific substrates [36]. Phosphorylation at S21 and S438 residues was significantly increased after GDC-0941 treatment, but SCH-772984 treatment had no effect. Phosphorylation at S21 residue represents a new modification of MST1, but phosphorylation at S438 residue has been previously linked to mTORC2 activity [19]. S438 was also reported to be essential for RASSF5 binding to MST1, suggesting the potential inhibition of this interaction after phosphorylation at S438 residue [37]. RASSF5 binding to MST1 moderates MST1 apoptotic signaling, suggesting the potential involvement of phosphorylation at S438 residue in the regulation of caspase during PI3K inhibition. We predicted PLK1 as a kinase that can phosphorylate MST1 at S21 residue and, NEK2 as an alternative kinase to mTORC2 that can phosphorylate MST1 at S438 residue (Figure 2B,C). Previous reports demonstrated the direct interaction of MST1/2 with PLK1 and NEK2, suggesting the likely phosphorylation of MST1 by these two kinases [38,39]. An increase in phosphorylation at S21 and S438 residues correlates with an increase in MEK1/2 activity, and the regulation of PLK1 and NEK2 signaling by MEK1/2 has been previously described [40,41]. Since the position of S21 residue is highly conserved in evolution, it could represent a crucial regulator of the MST1 protein functions. The phosphorylation levels at S40, T340, and S410 residues were significantly decreased only after SCH-772984 treatment which can possibly represent the inhibition of residual ERK1/2 activity or SCH-772984 off-target activity [42]. No information is available on the kinases responsible for this type of phosphorylation, but it was demonstrated that phosphorylation at T340 residue was important for MOB1 binding to MST1 [36]. Finally, phosphorylation at S414 residue was unaffected by GDC-0941, decreased by SCH-772984, and strongly decreased by a combination of GDC-0941 and SCH-772984. We predicted GSK3B as a kinase involved in MST1 phosphorylation at S414 residue (Figure 2D). The regulation of GSK3B signaling by P90RSK has been previously reported as well as regulation of the Hippo signaling by GSK3B [43,44]. It is well known that AKT, ERK, and P90RSK kinases inhibit GSK3B activity, which correlates with the insignificant effect of the GDC-0941 inhibitor on the phosphorylation levels of S414 residue and small decrease in the phosphorylation levels of S414 residue after treatment with SCH-772984. However, a strong inhibition of the phosphorylation levels at S414 residue after treatment with a combination of the PI3K and ERK1/2 inhibitors remains elusive. Moreover, we identified the significant enrichment of PLK1 and NEK2 peptides in a trypsin-digested mixture of proteins, which were co-precipitated with MST1 after GDC-0941 treatment (Appendix A). 

Several studies have highlighted the interplay between the activation of MST1 and cellular caspases, but this interplay shows a bidirectional character. Several studies have demonstrated the caspase-dependent proteolytic activation of MST1, whereas several others have shown the caspase-independent proteolytic activation of MST1 and the MST1-dependent activation of caspases. To clarify this Hippo–caspase concatenate, we determined the activity of caspase-1, -3, -6, -7, and -9 using colorimetric substrates specific for individual caspases. We observed a weak increase in the activity of caspase-6/-9 and a significant increase in the activity of caspase-3/-7 after 2 h of GDC-0941 treatment (Figure 3A). Using immunoblotting, we confirmed the proteolytic activation of caspases-3, -6, -7, and -8 (Figure 4) as a decrease in the levels of the non-cleaved forms together with an increase in the levels of the cleaved forms of caspase-3, -7, and -8. We could not detect the cleaved forms of caspase-6, suggesting that other mechanisms regulate the caspase-6 levels and activity in Jurkat T cells. Executive caspase-3/-7 have been previously reported to be responsible for MST1 cleavage during apoptosis. Therefore, we inhibited their activity using the Z-DEVD-FMK inhibitor, which covalently modifies the catalytic site of these caspases, and then monitored the effect of this irreversible inhibition on the proteolytic activation of MST1. We completely blocked the activation and activity of caspase-3/-7 after GDC-0941 treatment by pre-treating cells with 2 µmol/L Z-DEVD-FMK (Figure 3B and 4); however, we detected only a moderate decrease in the levels of the cleaved form of MST1 together with a decrease in the levels of the full-length form of MST1 (Figure 5A). These results indicate the existence of a mechanism alternative to MST1 proteolytic cleavage by caspase-3/-7 during GDC-0941 treatment. Pre-treatment of cells with 2 µmol/L Z-DEVD-FMK also completely blocked the activity of caspase-6/-9 during GDC-0941 treatment (Figure 3B), but this concentration was insufficient to completely block the proteolytic activation of caspase-8 (Figure 4). The proteolytic activation of caspase-8 and MST1 was completely blocked by the general inhibitor Z-VAD-FMK at 2 µmol/L concentration but a decrease in the levels of the full-length form of MST1 was still detected (Figure 5A). Moreover, the Z-VAD-FMK inhibitor at 1 µmol/L concentration completely blocked the activity of caspase-3/-7 (Figure 3B) but was also insufficient to fully block MST1 proteolytic cleavage (Figure 5A). These results demonstrate the potential crosstalk between MST1 and caspase-8 during GDC-0941 treatment in Jurkat T cells. To test the potential impact of MST1 on the activation of the extrinsic apoptotic pathway during GDC-0941 treatment, we inhibited MST1 levels using transfection with MST1-specific siRNA and monitored the proteolytic activation of caspase-3, -7, and -8 by immunoblotting. We confirmed a significant decrease in the levels of the cleaved forms of these caspases after MST1 inhibition together with a significant decrease in the activity of caspase-3/-7, as determined using the Ac-DEVD-pNA colorimetric substrate (Figure 5B,C). These results illustrate MST1 as an ERK-dependent activator of caspase-8 and the extrinsic apoptotic pathway in Jurkat T cells. The activation of the caspase-8-caspase-3/7 pathway by MST1 enhanced the proteolytic activation of MST1. This mechanism forms a positive feed-back loop in MST1 signaling, providing amplification of the apoptotic signal in Jurkat T cells (Figure 6). Detailed characterization of the mechanisms responsible for the activation of MST1 represents the future direction in Hippo signaling pathway research. 

## 4. Materials and Methods

### 4.1. Cell Line and Treatment

The T-cell lymphoma-derived Jurkat T cell line, clone E6.1, was obtained from the ATCC collection (ATCC, Manassas, VA, USA) and cultured at 37 °C under 5% CO_2_ in RPMI1640 medium supplemented with L-glutamine (Lonza Group, Ltd., Basel, Switzerland), 10% fetal bovine serum (Gibco, Thermo Fisher Scientific, Waltham, MA, USA), 100-U/mL penicillin, and 100-µg/mL streptomycin (Thermo Fisher Scientific). The cells were seeded at a density of 2.5 × 10^5^ cells/mL in 15 mL of RPMI1640 medium, grown overnight in 75-cm^2^ cell culture flasks (TPP, Trasadingen, Switzerland) under standard cultivation conditions, and treated with GDC-0941 (Selleckchem, Houston, TX, USA) at a final concentration of 10 µmol/L or SCH-772984 (Selleckchem, Houston, TX, USA) at a final concentration of 1 µmol/L for 2 h. Alternatively, cells were pre-treated with SCH-772984 at a final concentration of 1 µmol/L concentration (effective concentration for Jurkat T cells according to the manufacturer) for 1 h and then treated with GDC-0941 at a final concentration of 10 µmol/L for 2 h (concentration inducing apoptosis in Jurkat T cells; [45]). GDC-0941 and SCH-772984 stock solutions (10 mmol/L in DMSO, kept at −80 °C) were diluted to appropriate concentrations in the culture medium. The cells treated with an equal volume of DMSO (Sigma-Aldrich, St. Louis, MO, USA) were used as a control. Cytosolic fractions were isolated using the Nuclear Extract Kit (Active Motif, La Hulpe, Belgium), and a bicinchoninic acid (BCA) assay was used to determine the total protein concentration in cell lysates (Thermo Fisher Scientific).

### 4.2. Immunoblotting

Precast gradient TGX gels (4–15%, Bio-Rad, Hecules, CA, USA) were loaded with 40 µg of proteins isolated from individual samples per lane. A Protean III apparatus (Bio-Rad, Hecules, CA, USA) with a constant voltage of 100 V was used to run the SDS-PAGE protein samples. Separated proteins were blotted onto a nitrocellulose membrane (Bio-Rad, Hecules, CA, USA) using a Trans-Blot™ SD Semi-Dry apparatus (Bio-Rad, Hecules, CA, USA). Protein blots were blocked for 1 h in TBS supplemented with 5% non-fat milk (Bio-Rad, Hecules, CA, USA) and 0.05% Tween-20 (Sigma-Aldrich). Membranes were washed with TBS containing 0.05% Tween-20 and incubated with the respective primary and secondary antibodies as per the manufacturer’s protocols. The following antibodies were used for immunostaining: MST1, AKT, p-AKT (S473), ERK1/2, p-ERK1/2 (T202/Y204), p-RAF (S338), p-MEK1/2 (S217/S221), p-P90RSK (S380), p-MSK1 (T581), caspase-3, caspase-6, caspase-7, and caspase-8 antibodies (Cell Signaling Technology, Danvers, MA, USA); p-P70S6K (S424) and actin antibodies (Santa Cruz Biotechnology, Dallas, TX, USA); and HRP-conjugated secondary antibodies against rabbit and goat IgG (Santa Cruz Biotechnology, Dallas, TX, USA). WestPico ECL substrate (Thermo Fisher Scientific) and ChemiDoc CCD system (Bio-Rad, Hercules, CA, USA) were used for chemiluminescence signal detection and visualization. A representative image of each immunoblot was selected from at least three independent experiments.

### 4.3. Kinetic Measurements of Proteolytic Activity against Individual Substrates

Caspase activities in cytosolic fractions were determined using specific substrates conjugated with the *p*-nitroanilide group (pNA). Each reaction contained 100 µg of proteins from cytosolic fractions resuspended in 50 µL of hypotonic buffer, 50 µL of 2× caspase assay buffer (20-mmol/L PIPES pH 7.4, 100-mmol/L NaCl, 1-mmol/L EDTA, 0.1% CHAPS, 10% sucrose, and 10-mmol/L dithiothreitol; Sigma-Aldrich), and 5 µL of specific substrate (10 mmol/L in DMSO, kept at −20 °C). The following substrates were used: Ac-LEHD-pNA (Sigma-Aldrich), Ac-VEID-pNA, Ac-YVAD-pNA, and Ac-DEVD-pNA (Santa Cruz Biotechnology, Dallas, TX, USA). Samples were pre-incubated in 96-well plates at 37 °C for 15 min, and absorbance at 405-nm wavelength (A_405_) was recorded using a plate reader (60 min, read period 60 s; BioTek, Winooski, VT, USA). The background was subtracted from all A_405_ values, and the resulting A_405_ values from two technical replicates (mean only) were plotted in a time-dependent manner. The best-fit regression line was then superimposed using GraphPad software. Each graph shows the results of one representative measurement in a technical duplicate selected from at least three independent experiments.

### 4.4. Inhibition of Caspase Activity

After overnight cultivation, cells were pre-treated with the selected concentrations of individual inhibitors conjugated with the fluoromethylketone group (FMK) instead of the pNA group. Stock solutions (10 mmol/L in DMSO, kept at −20 °C) of the following inhibitors were used: Z-DEVD-FMK and Z-VAD-FMK (Santa Cruz Biotechnology, Dallas, TX, USA). Cells were pre-treated with individual inhibitors for 1 h, treated with GDC-0941 for 2 h, and processed as per the standard protocols.

### 4.5. RNA Interference

MST1 expression was inhibited by transfection with a pool of four MST1-specific siRNAs (Thermo Fisher Scientific) into Jurkat T cells using the DharmaFECT4 reagent (Thermo Fisher Scientific). A non-targeting pool of four individual siRNAs was used as a control. MST1 protein levels were determined by immunoblotting 72 h after transfection. Transfected cells were treated and processed as per the standard protocols.

### 4.6. MST1 Antibody Coupling to the Dynabeads M-270 Epoxy

MST1 antibody (monoclonal rabbit IgG; Sigma-Aldrich) was used for co-immunoprecipitation experiments using the Dynabeads Co-Immunoprecipitation Kit (Thermo Fisher Scientific). For this purpose, 5 µg of MST1 antibody per mg of Dynabeads (10 mg of beads in total) was coupled as per the manufacturer’s protocol.

### 4.7. Immunoprecipitation of Native MST1 Protein

Cells were seeded at a density of 7 × 10^5^ cells/mL in 150 mL of RPMI1640 medium in 300-cm^2^ culture flasks (TPP, Trasadingen, Switzerland) and cultivated under standard conditions for 1 h. Cells were then treated with the selected compounds as per the standard protocol. Cells were collected by centrifuging, washed once with PBS, and lysed by the detergent lysis method. Cells were resuspended in a 1:9 ratio of cell mass to extraction buffer with protease inhibitors (Halt Protease Inhibitors, Thermo Fisher Scientific), incubated on ice for 15 min, and then centrifuged at 2600× *g* for 5 min. The obtained supernatant was immediately used for co-IP. After co-IP, the precipitated proteins were eluted in 1000 µL of HPH EB buffer. We saved 100 µL of eluates for the MS identification of co-precipitated proteins and separated lyophilized eluates using SDS-PAGE followed by Coomassie staining for visualization.

### 4.8. In-Gel Trypsin Digestion of MST1

Eluates from immunoprecipitation were precipitated by adding four volumes of ice-cold acetone, kept at −20 °C for 30 min, and centrifuged at 16,000× *g* and 4 °C for 20 min. The supernatant was removed, and cell pellets were resuspended in 100 mM TEAB containing 2% SDC, followed by boiling at 95 °C for 5 min. Cysteines were reduced with TCEP at a final concentration of 5 mM (60 °C for 60 min) and blocked with MMTS at a final concentration of 10 mM (room temperature for 10 min). Samples were digested with trypsin (trypsin:protein ratio, 1:20) at 37 °C overnight. After digestion, samples were acidified with TFA at a final concentration of 1%. SDC was removed by extraction with ethyl acetate and the peptides were desalted in a Michrom C18 column. Dried peptides were resuspended in 25 µL of water containing 2% acetonitrile (ACN) and 0.1% trifluoroacetic acid. For analysis, 12 µL of sample was injected [46].

### 4.9. In-Solution Trypsin Digestion of Precipitated Proteins

Individual bands containing proteins of interest were excised from the Coomassie-stained SDS-PAGE gel using a razor blade and cut into small pieces (approximately 1 mm × 1 mm). Bands were destained by sonication for 30 min in 50% ACN and 50 mM ammonium bicarbonate (ABC). After destaining, the solution was removed and gels were dried in ACN. Disulfide bonds were reduced using 10 mm DTT in 100 mM ABC, at 60 °C, for 30 min. Subsequently, samples were re-dried with ACN, and free cysteine residues were blocked using 55 mM iodoacetamide in 100 mM ABC in the dark, at room temperature for 10 min. Samples were dried thoroughly, and digestion buffer (10% ACN, 40 mM ABC, and 13-ng/µL trypsin) was added to cover gel pieces. Proteins were digested at 37 °C overnight. After digestion, 150 µL of 50% ACN with 0.5% formic acid was added, followed by sonication for 30 min. The supernatant containing peptides was added to a new microcentrifuge tube, another 150 µL of elution solution was added to the supernatant, and this solution was sonicated for 30 min. The solution was then removed, combined with the previous solution, and dried using Speedvac. Dried peptides were reconstituted in 2% ACN with 0.1% TFA and injected into Ultimate 3000 Nano LC coupled to Orbitrap Fusion.

### 4.10. NanoLC–MS^2^ Analysis

A nano reversed-phase column (EASY-Spray column, 50-cm × 75-µm inner diameter, PepMap C18, 2-µm particle size, 100-Å pore size) was used for LC–MS analysis. Mobile phase buffer A was composed of water and 0.1% formic acid. Mobile phase buffer B was composed of ACN and 0.1% formic acid. Samples were loaded onto the trap column (Acclaim PepMap300, C18, 300 µm × 5 mm inner diameter, 5-µm particle size, 300-Å pore size) at a flow rate of 15 μL/min. Loading buffer was composed of water, 2% ACN, and 0.1% trifluoroacetic acid. Peptides were eluted with buffer B gradient from 4% to 35% over 60 min at a flow rate of 300 nL/min. Eluting peptide cations were converted to gas-phase ions by electrospray ionization and analyzed on a Thermo Orbitrap Fusion (Q-OT-qIT, Thermo Fisher Scientific). Survey scans of peptide precursors from 350 to 1400 *m*/*z* were performed at 120K resolution (at 200 *m*/*z*) with a 5 × 10^5^ ion count target. Tandem MS was performed by isolation at 1.5 Th with the quadrupole, HCD fragmentation with normalized collision energy of 30, and rapid scan MS analysis in the ion trap. The MS^2^ ion count target was set to 10^4^, and the max injection time was set to 35 ms. Precursors with a charge state of 2–6 were sampled for MS^2^. The dynamic exclusion duration was set to 45 s with a 10 ppm tolerance around the selected precursor and its isotopes. Monoisotopic precursor selection was turned on. The instrument was run in the top speed mode with 2 s cycles.

### 4.11. MS Data Analysis

All data were analyzed and quantified using MaxQuant software (version 1.6.1.0) [47]. The false discovery rate was set to 1% for both proteins and peptides, and a minimum length of seven amino acids was specified. The Andromeda search engine was used for the MS/MS spectra search against the Human SwissProt database (downloaded from UniProt on September 2017, containing 20,142 entries). Enzyme specificity was set as C-terminal to Arg and Lys, also allowing cleavage at proline bonds and a maximum of two missed cleavages. Carbamidomethylation of cysteine was selected as fixed modification, and N-terminal protein acetylation, methionine oxidation, and serine/threonine phosphorylation were selected as variable modifications. Data analysis was performed using Perseus 1.5.2.4 software and Skyline 4.1.0.18169 [48].

## Figures and Tables

**Figure 1 cells-08-00191-f001:**
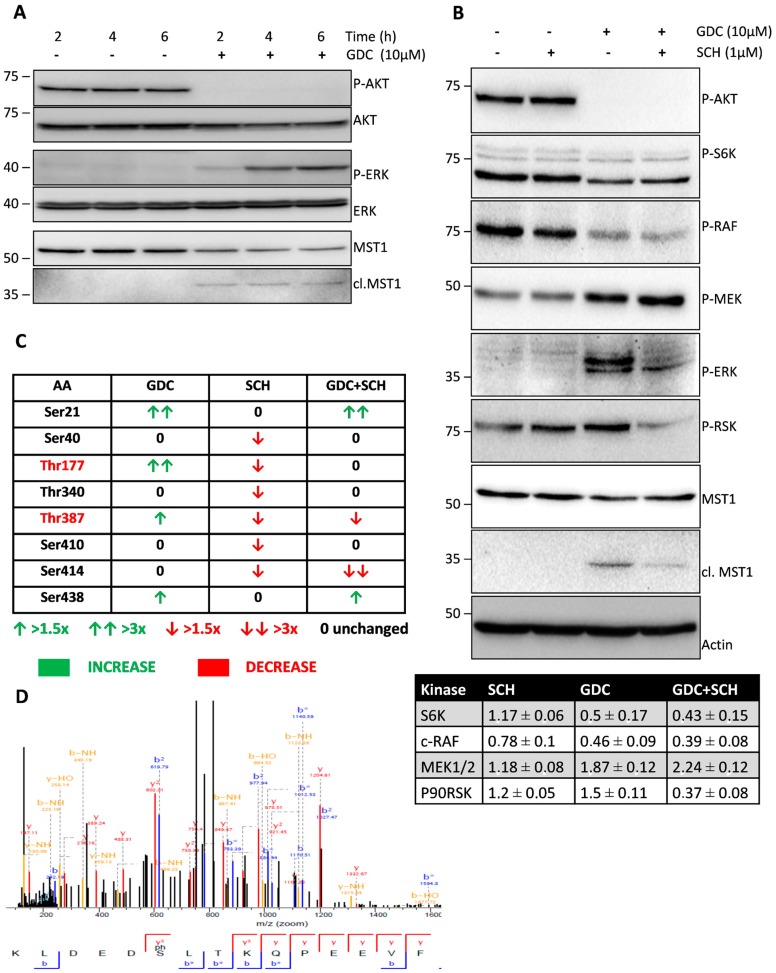
Effect of GDC-0941, SCH-772984, and their combination on the status of selected kinases. (**A**) Cells were treated with GDC-0941 for the time periods shown. Levels of phosphorylated and total kinases were determined by immunoblotting. (**B**) Cells were treated with GDC-0941, SCH-772984, and their combination for 2 h. Effect on the phosphorylation levels of individual kinases and MST1 cleavage was detected by immunoblotting. Actin was used as a loading control. Relative changes in the phosphorylation levels of selected kinases normalized to control are shown in the table. Mean of three independent experiments ± SD. (**C**) Phosphorylation at individual AA residues of MST1 was quantified by LC–MS/MS analysis. Relative changes in the phosphorylation levels against control cells are demonstrated by arrows. Residues phosphorylated in an ERK1/2-dependent manner are shown in red color. (**D**) The MS/MS spectrum of a novel phosphorylation site of MST1 at S21 residue.

**Figure 2 cells-08-00191-f002:**
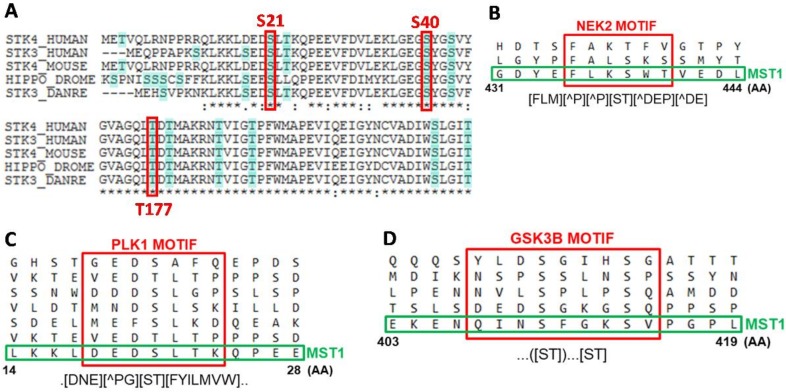
Bioinformatic analysis of MST1 protein residues identified to be phosphorylated. (**A**) Conservation of positions of S21, S40, and T177 residues in STK3/STK4 sequences of humans, mice, flies, and fish. (**B**–**D**) Sequences of motifs recognized by individual kinases around S438, S21 and S414 residues according to the ELM resource. Common motifs are shown in the syntax of Regular Expression Parser and Visualizer (https://github.com/CJex/regulex). (AA) symbol represents AA constrained in the motif, (AA) symbol represents AA excluded from the motif and dot symbol represents random AA.

**Figure 3 cells-08-00191-f003:**
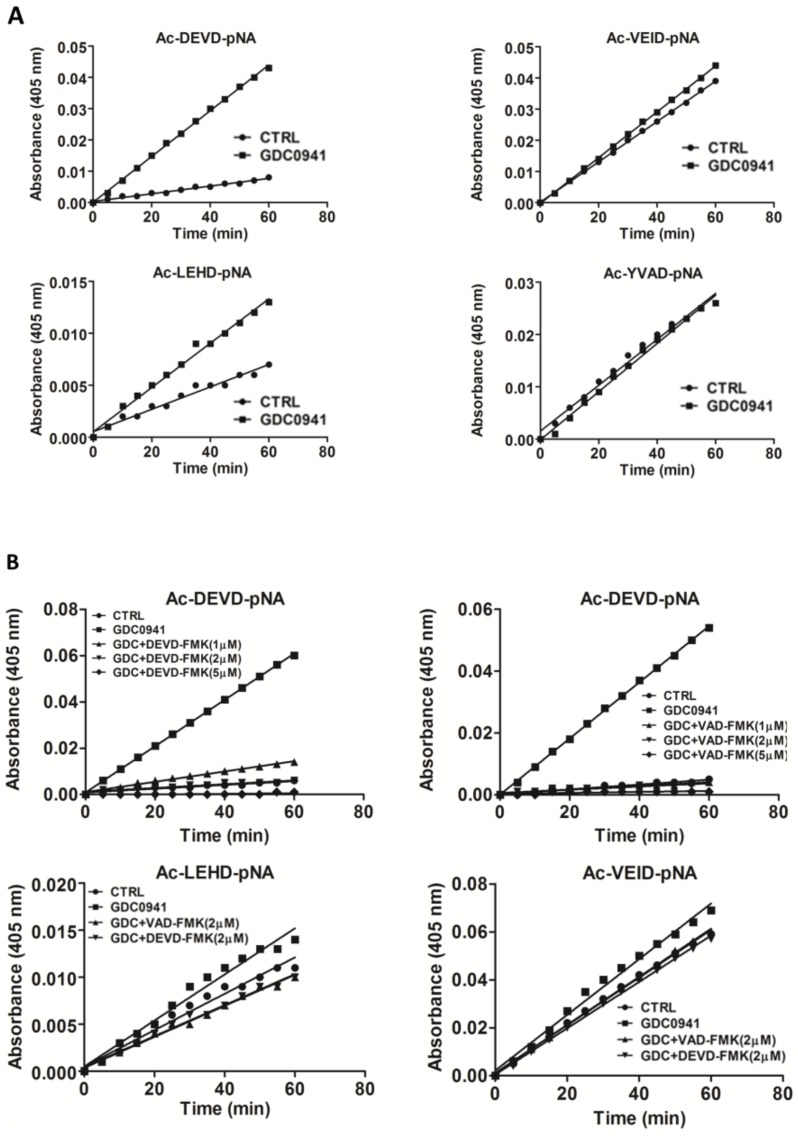
Effect of selected protease inhibitors on the activity of individual caspases during GDC-0941 treatment. (**A**) Proteolytic activities in cell lysates toward individual substrates were determined 2 h after GDC-0941 treatment. (**B**) Effect of pre-treatment of cells with selected concentrations of the Z-DEVD-FMK and Z-VAD-FMK protease inhibitors on the proteolytic activity in cell lysates. Activity was determined 2 h after GDC-0941 treatment.

**Figure 4 cells-08-00191-f004:**
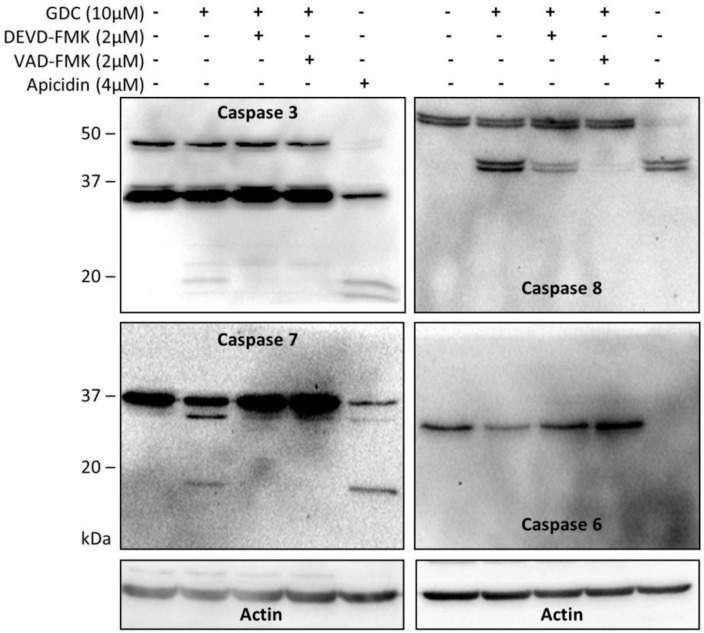
Effects of selected protease inhibitors on the proteolytic activation of individual caspases during GDC-0941 treatment. The proteolytic activation of caspases in control cells, cells treated with GDC-0941 for 2 h, and cells pre-treated with the Z-DEVD-FMK and Z-VAD-FMK protease inhibitors before GDC-0941 treatment was detected by immunoblotting. Cells treated with apicidin for 24 h were used as a positive control.

**Figure 5 cells-08-00191-f005:**
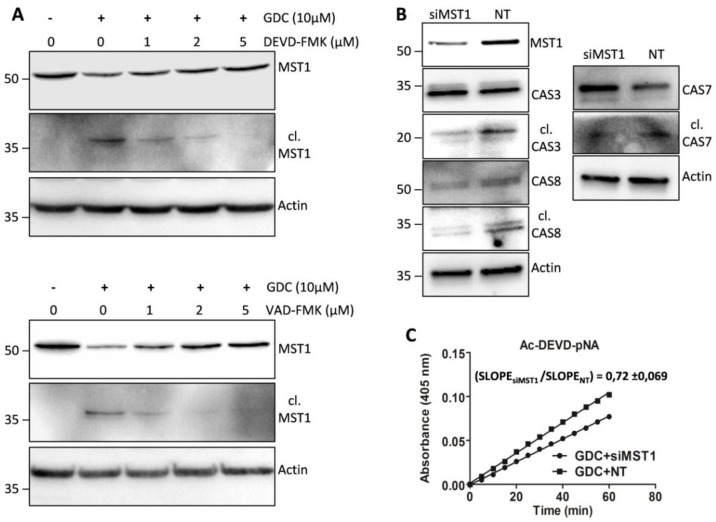
Effect of protease inhibitors on the proteolytic activation of MST1 and regulation of selected caspases by MST1 levels after GDC-0941 treatment. (**A**) The proteolytic activation of MST1 in non-treated cells and cells pre-treated with selected concentrations of Z-DEVD-FMK and Z-VAD-FMK was detected by immunoblotting after 2 h of GDC-0941 treatment. (**B**) The activation of caspases in cells transfected with siRNAs against MST1 (siMST1) and non-target (NT) siRNAs. The activation of individual caspases was detected by immunoblotting after 2 h of GDC-0941 treatment. (**C**) Proteolytic activity against the Ac-DEVD-pNA substrate in cells transfected with siMST1 and NT siRNAs. Activity was determined in cell lysates after 2 h of GDC-0941 treatment. Slope ratio was determined in three independent experiments.

**Figure 6 cells-08-00191-f006:**
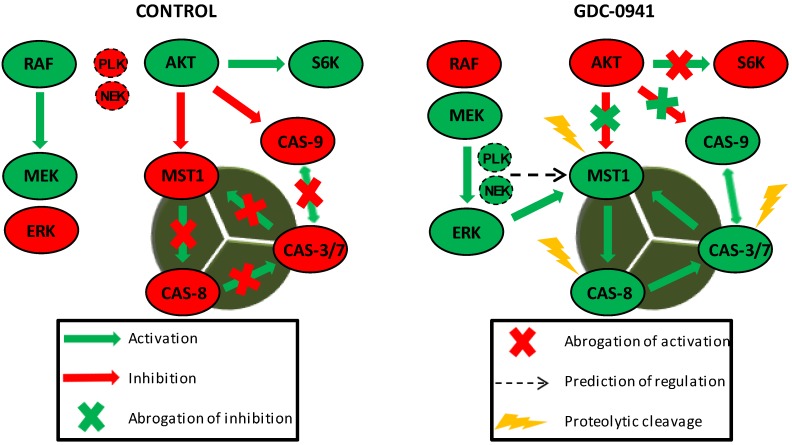
MST1 acts as a regulator of the extrinsic apoptotic pathway in Jurkat T cells. PI3K signaling blockade by GDC-0941 resulted in the abrogation of AKT kinase phosphorylation at S473 and significantly attenuated the phosphorylation of P70S6K and c-RAF kinase. This led to a paradoxical increase in the phosphorylation of MEK1/2 at S217/S221 and ERK1/2 at T202/Y204 followed by the activation of MEK-ERK signaling. Active ERK1/2 signaling stimulated the proteolytic activation of MST1 which was insensitive to the experimentally determined concentration of the Z-DEVD-FMK inhibitor. At selected concentrations, Z-DEVD-FMK effectively blocked the activity of caspase-3, -7 and -9 but was insufficient to block the proteolytic activation of caspase-8, suggesting the potential activation of MST1 by caspase-8. On the other hand, the inhibition of MST1 levels by siRNA significantly reduced the proteolytic activation of caspase-3, -7 and -8, rendering MST1 a potential regulator of caspase-8 activity in Jurkat T cells. Moreover, activity of caspase-3/-7 potentiates the proteolytic activation of MST1, hence we presume the existence of a positive feed-back loop orchestrated by MST1 to amplify the apoptotic signal in Jurkat T cells. PLK1 and NEK2 were predicted as potential regulators of MST1 activity during PI3K signaling blockade. Cleavage of caspase-9 was not determined in this study.

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
