# Peer review of "The MEK-ERK-MST1 Axis Potentiates the Activation of the Extrinsic Apoptotic Pathway during GDC-0941 Treatment in Jurkat T Cells"

_cells, 2019, doi:10.3390/cells8020191_

Round 1

Reviewer 1 Report

While the authors have answered my previous comments, the changes to the manuscript have been limited to removal of data rather than strenghthening of the conclusions. I certainly do believe that experiments performed in a single most commonly used lymphoid cell line lilke Jurkat to make sweeping statement on ``acute lymphoblastic leukemia`` is not acceptable. The only way this story can be acceptable is by 

changing the title and conclusions to '' ....GDC-0941 treatment in jurkat T cells''

page 223:''....highlights MST1 kinase as a novel regulator of caspase activity and apoptosis in ALL cells''. Remove ''novel'' as MST1 is an already established regulator of apoptosis

Line 225: ''MST1 acts as a caspase-independent activator of extrinsic apoptotic pathway...`` This is not true and no evidence is provided for the fact that cell death in this system in caspase independent. Apoptotsis is per se caspase-dependent death.

line 108-9: change to ``...phosphirylated and total kinases were.....``

Author Response

The only way this story can be acceptable is by changing the title and conclusions to '' ....GDC-0941 treatment in jurkat T cells''

We changed name of model cell line used here to Jurkat T cells/Jurkat cells in whole manuscript.

page 223:''....highlights MST1 kinase as a novel regulator of caspase activity and apoptosis in ALL cells''. Remove ''novel'' as MST1 is an already established regulator of apoptosis

Sentence was changed according reviewer comment.

Line 225: ''MST1 acts as a caspase-independent activator of extrinsic apoptotic pathway...`` This is not true and no evidence is provided for the fact that cell death in this system in caspase independent. Apoptotsis is per se caspase-dependent death.

We changed this statement to ''MST1 acts as a regulator of extrinsic apoptotic pathway...``

line 108-9: change to ``...phosphirylated and total kinases were.....``

Sentence was changed according reviewer comment.

Reviewer 2 Report

In the paper by Jana Novakova et al., titled “MEK-ERK-MST1 axis potentiates activation of extrinsic apoptotic pathway during GDC-0941 treatment in acute lymphoblastic leukemia” authors present data supporting observation for apoptosis induction upon AKT inhibitor treatment and Mst1 kinase activation/cleavage. The outline and scientific merit is of interest for Cells readers and thus it is recommended for publication. However there are couple of unclarified points that need to be corrected prior publication.

It is unclear if GDC-0941 induces only extrinsic apoptosis not intrinsic apoptosis pathway as well. In the Figure 6 authors admit that in addition to caspase 8 also caspase 9 is likely to be activated by AKT inhibitor GDC-0941, however throughout the paper authors explain apoptosis as an extrinsic process dependent on caspase 8. And although there is a convincing evidence with Western Blot analysis and use of specific peptide to inhibit caspase 8, there is no additional data on caspase 9 and intrinsic apoptosis that could be activated due to AKT/mTOR1 inhibition. Additional Western Blots with either Mst1 siRNA and drug treatment with cytochrome c release or staining for caspase 9 after drug treatment would complement the study. If the authors are unable to provide additional data on intrinsic pathway of apoptosis induced by GDC-0941 it needs to be clarified in the title and results section.

Residues on Mst1 protein recognized and mapped based on MS analysis provide some valid information on kinases that are activated during AKT inhibition such as MEK/ERK, and thus may determine Mst1 kinase activation by caspases. In particular, S21 site was identified as a new site to be phosphorylated on Mst1, most likely by ERK1/2 kinase. Still S21 site on Mst1 was not validated with overexpression of phospho-mutant versions of Mst1 protein and drug treatment. It is unclear whether S21 is needed for efficient apoptosis induction.

Authors use two specific kinase inhibitors, GDC-0941 to inhibit AKT and SCH-772984 to inhibit ERK. It remains unknown if both of these inhibitors induce greater apoptosis in combination and how it relates to ERK activation upon AKT inhibition. Thus, it would be helpful to provide some information if upon inhibition of both ERK and AKT kinases there is a greater apoptosis in Jurkat cells. On the other hand if there is inhibition of apoptosis induced by AKT inhibitor with concomitant ERK inhibition, it suggests involvement of ERK kinase in apoptosis induction upon GDC treatment.

In Figure 5C graph presenting read-out of caspase activity, authors say that siMist1 caused a “significant difference in apoptosis induction as compared to NT-siRNA treated cells” without actually providing any statistical analysis for this assay.

Author Response

It is unclear if GDC-0941 induces only extrinsic apoptosis not intrinsic  apoptosis pathway as well. In the Figure 6 authors admit that in addition to caspase 8 also caspase 9 is likely to be activated by AKT inhibitor GDC-0941, however throughout the paper authors explain apoptosis as an extrinsic process dependent on caspase 8. And although there is a convincing evidence with Western Blot analysis and use of specific peptide to inhibit caspase 8, there is no additional data on caspase 9 and intrinsic apoptosis that could be activated due to AKT/mTOR1 inhibition. Additional Western Blots with either Mst1 siRNA and drug treatment with cytochrome c release or staining for caspase 9 after drug treatment would complement the study. If the authors are unable to provide additional data on intrinsic pathway of apoptosis induced by GDC-0941 it needs to be clarified in the title and results section.

We have demonstrated increase in proteolytic activity against Ac-LEHD-pNA substrate, which is specific for caspase-9 during GDC-0941 treatment (Figure 3A) and increase in LEHD-pNA proteolytic activity abolishment by Z-DEVD-FMK inhibitor (Figure 3B) which is specific for executive caspases. These observations suggest activation of caspase-9 as a result of activity of executive caspases (e.g. activation of caspase-9 by tBID). We have depicted this fact in graphical abstract. 

Residues on Mst1 protein recognized and mapped based on MS analysis provide some valid information on kinases that are activated during AKT inhibition such as MEK/ERK, and thus may determine Mst1 kinase activation by caspases. In particular, S21 site was identified as a new site to be phosphorylated on Mst1, most likely by ERK1/2 kinase. Still S21 site on Mst1 was not validated with overexpression of phospho-mutant versions of Mst1 protein and drug treatment. It is unclear whether S21 is needed for efficient apoptosis induction.

We have demonstrated phosphorylation of S21 residue as an ERK-independent process and predicted PLK1 as a responsible kinase. Due to conservation of this residue in evolution we presumed potential function of S21 in the regulation of MST1 activity. SCH-772984 had no effect on increase in S21 phosphorylation levels but inhibited MST1 proteolytic activation after GDC-0941 treatment. Hence, it seems that S21 has no effect on MST1 proteolytic activation after GDC-0941 treatment. Functional characterization of S21 represents interesting topic for separate study in the future.  

Authors use two specific kinase inhibitors, GDC-0941 to inhibit AKT and SCH-772984 to inhibit ERK. It remains unknown if both of these inhibitors induce greater apoptosis in combination and how it relates to ERK activation upon AKT inhibition. Thus, it would be helpful to provide some information if upon inhibition of both ERK and AKT kinases there is a greater apoptosis in Jurkat cells. On the other hand if there is inhibition of apoptosis induced by AKT inhibitor with concomitant ERK inhibition, it suggests involvement of ERK kinase in apoptosis induction upon GDC treatment.

We focused primarily on mechanisms involved in MST1 proteolytic activation induced by GDC-0941, hence we did not determine apoptosis in general. SCH-772984 was used only for identification of AA residues of MST1 phosphorylated in an ERK-dependent manner and demonstration of positive effect of ERK1/2 on MST1 proteolytic activation during GDC-0941 treatment. Induction of apoptosis through ERK signaling was reported previously however effect on MST1 proteolytic activation was described for the first time in this study. We presume that degree of MST1 cleavage positively correlates with extent of apoptosis.

In Figure 5C graph presenting read-out of caspase activity, authors say that siMist1 caused a “significant difference in apoptosis induction as compared to NT-siRNA treated cells” without actually providing any statistical analysis for this assay.

The slope quantifies the steepness of the line fitted by linear regression to experimental data. In enzyme kinetics slope corresponds to relative enzymatic activity. Hence, we provided mean value (±SD) of ratios of slopes calculated from regression lines determined in three independent experiments (siMST1/NT ratio). Result shows average proteolytic activity against DEVD substrate after MST1 inhibition as 72% of control cells (NT).

Round 2

Reviewer 1 Report

Authors have now mad the necessary changes which I asked for in the last round of revision.

Reviewer 2 Report

Authors responded to my comments with small edits throughout the text of this article, change in a Figure legent, and thus article in its current form is acceptable for publication.

This manuscript is a resubmission of an earlier submission. The following is a list of the peer review reports and author responses from that submission.

Round 1

Reviewer 1 Report

It contains valuable data bout need to improve.

First of all, please reorganize figure number accoridng to result section.

Please add some summarizing figure for mechanism proposed.

GDC0941 and SCH-772984 are chemical inhibitor. They might has  inhibitory effects against other kinases. Please discuss about these things.

Discussion part  seem to be too long.

Please compare these reuslts with those of gene silencing of siPI3K or dominant negative of PI3K in cells reported. What are the difference and similarity ?

In title what is the meaning of "control". Does it mean to promote or inhibit?

Please use direct word.

Is the concentration of chemical used valid? Please add reference  for justification of use.

Several phosphorylation site has consensus sequece. Please discuss the phosphorylation site with these consensus sequences. 

Reviewer 2 Report

In the manuscript by Novakova et al, authors investigate MST1 signaling during PI3K-inhibition in leukemic cells and propose that there is a feed forward network which involves ERK1/2 activity, MST1 and CASP8. I find this study rather confusing and premature. Many of the statements and assumptions are not really convincing. The major limitations of the study are

1.       The entire work uses single cell line and uses single inhibitor  against kinase targets- making it really hard to provide supportive data for the very general title

2.       Caspase-8 is certainly an initiator caspase and will be involved in most of the extrinsic cell death pathways. MST1 is known to be a positive regulator of apoptosis and hence caspase activation (as observed by siMST1 in Fig 5B). It is hard to understand the take home message of most of the experiments. For example: Fig4 just shows that caspase inhibitors inhibit caspases!

3.       The dose titration of caspase inhibitors and the conclusions made from those in Fig 5A is really not convincing

4.       Most of the western immunoblotting will be only convincing, if the authors provide quantification of the triplicate experiments (eg. in Fig 1B, there is no total MSK/RSK, so at least showing that the minor effects at the phospho-level are reproducible is key)

5.       The assumption of kinase identity by MS and speculations (Fig 1c-d) and further interaction verification just by MS analysis (Fig. 2) is also not acceptable. Normally MS data will need verification by other means or at least needs multiple MS assays to confirm the findings.

Minor points:

       page 3, line 111: authors suggest ``PI3K-AKT-RAF pathway``. I don’t think such a pathway exist. Even though PI3K-àRAF link is blurred, Akt is not normally part of this link

  Authors state that since PI3K inhibitor-induced ERK1/2 activation is not mediated by RAF, it is non canonical. This could be an after effect of cell death, mediated by STK1 itself or by other MAP3Ks.